# A Positive Feedback DNA-PK/MYT1L-CXCR1-ERK1/2 Proliferative Signaling Loop in Glioblastoma

**DOI:** 10.3390/ijms26094398

**Published:** 2025-05-06

**Authors:** Bo Wang, Dongping Li, Yaroslav Ilnytskyy, Levon M. Khachigian, Nuanying Zhong, Rocio Rodriguez-Juarez, Igor Kovalchuk, Olga Kovalchuk

**Affiliations:** 1Department of Biological Sciences, University of Lethbridge, Lethbridge, AB T1K 3M4, Canada; bo.wang5@uleth.ca (B.W.); dongping.li@uleth.ca (D.L.); yaroslav.ilyntskyy@uleth.ca (Y.I.); zhongnuanying@gmail.com (N.Z.); rodrrc@uleth.ca (R.R.-J.); 2Vascular Biology and Translational Research, School of Medical Sciences, University of New South Wales, Sydney, NSW 2052, Australia; l.khachigian@unsw.edu.au

**Keywords:** MYT1L, glioblastoma, DNA-PK, proliferative signaling loop, CXCR1

## Abstract

Glioblastoma is the most common primary brain tumor in adults. Our previous studies revealed a functional interplay of myelin transcription factor 1-like (MYT1L) with the DNA-dependent protein kinase (DNA-PK) in the regulation of p21 transcription. However, the contributing role of this functional interplay in glioblastoma remains largely unknown. Here, we used cell lines with normal DNA-PK (HEK293 and M059K) or deficient DNA-PK (M059J) as a model system to demonstrate the importance of the DNA-PK-dependent activation of MYT1L in controlling the transcription of CXC chemokine receptor 1 (CXCR1) in a positive-feedback proliferative signaling loop in glioblastoma with numerous conventional techniques. In normal DNA-PK cells, MYT1L acted as an oncogene by promoting cell proliferation, inhibiting apoptosis, and shortening a cell cycle S phase. However, in DNA-PK-deficient cells, MYT1L functioned as a tumor suppressor by inhibiting cell proliferation and inducing a G1 arrest. The enforced expression of MYT1L promoted CXCR1 transcription in DNA-PK-normal cells but attenuated transcription in DNA-PK-deficient cells. Bioinformatics analysis predicted a MYT1L-binding sequence at the CXCR1 promoter. The functional dependence of MYT1L on DNA-PK in CXCR1 transcription was validated by luciferase assay. Although the expression of CXCR1 was lower in M059J cells as compared to M059K cells, it was higher than in normal brain tissue. The CXCR1 ligands interleukin 8 (IL-8) and GRO protein alpha (GROα) expressed in M059J and M059K cells may signal through the extracellular signal-regulated kinase 1/2 (ERK1/2) pathway that can be blocked by CXCR1 siRNA. Our findings demonstrate the existence of a positive feedback DNA-PK/MYT1L-CXCR1-ERK1/2 proliferation loop in glioblastoma cells that may represent a pharmacological target loop for therapeutic intervention.

## 1. Introduction

Glioblastoma (also known as glioblastoma multiforme) is the most aggressive brain tumor and the most common primary brain malignancy in adults [1,2]. In Canada, between 2013 and 2017, the age-standardized incidence rate for glioblastoma has been reported to range from 0.22 per 100,000 (age group: 0 to 19 years) to 13.4 per 100,000 (age group ≥ 65 years) [2]. Patients with glioblastoma generally have a poor prognosis, with a two-year survival rate of 4.9% [2]. Although in the last two decades great advances have been made in our understanding of the genetic, epigenetic, and molecular events that contribute to the initiation and progression of glioblastoma, the mechanisms of glioblastoma development are not completely clear.

A range of dysfunctional signaling pathways have been linked to glioblastoma, including autophagy [3], metabolic [4], and activation of autocrine and/or paracrine growth factors and their receptors pathways. Among the growth factor/receptor pathways that are thought to play a role, those involving the epidermal growth factor receptor (EGFR, also known as HER1) [5,6], the platelet-derived growth factor (PDGF) [6,7,8,9], and the vascular endothelial growth factor (VEGF) [10] are frequently activated in glioblastoma. EGFR gene amplification is the most common genetic alteration associated with malignant glioma [11], accounting for 50% of this disease. Furthermore, EGFR gene rearrangement has been shown in a majority of glioblastoma cases (70–90%) with EGFR overexpression [12,13]. The most common form of the mutant EGFR in glioblastoma is EGFRvIII [13,14], which harbors an in-frame deletion of exon 2–7, resulting in truncation of the extracellular domain. Eventually, activation of EGFRvIII leads to the loss of dependence on ligand binding, displaying sustained activation. Although amplification of PDGF receptor genes is not an event as common as EGFR amplification in glioblastoma [15], PDGF receptors and their ligands are frequently up-regulated in glioblastoma cell lines and tissue samples [7,8,16,17,18]. Activation of PDGF receptor β (PDGFRB) enhances the self-renewal of glioma stem cells [19], whereas depletion of its ligand PDGFB completely attenuates the initiating capacity in this population [20]. Furthermore, the key role of PDGFB signaling in the initiation of glioblastoma has been validated in mouse models by an injection of a murine retroviral vector expressing PDGFB [21]. In this mouse model, the authors found that the retroviral vector expressing PDGFB primarily drives the development of glioblastoma, which consistently expresses nestin protein, a well-known biomarker for neuroglial progenitor cells, indicating that activation of PDGFB signaling may be an initial or early event in neuro-oncogenesis. Vascular homeostasis is tightly regulated by a balance between pro- and anti-angiogenic factors. VEGF is a primary pro-angiogenic factor that is thought to drive new blood vessel formation in glioblastoma [22]. VEGF and VEGF receptors (VEGFR) are up-regulated in glioblastoma cell lines and tumor tissues [23,24,25], while VEGF expression levels correlate with the grade of glioma. Furthermore, the VEGF receptor is overexpressed in tumor endothelial cells in vivo [23]. EGFR, PDGFR, and VEGFR are all transmembrane tyrosine kinase receptors that are generally activated by high-affinity ligand binding, resulting in signaling via mitogen-activated protein kinase (MAPK), phosphatidylinositol 3-kinase (PI3K)/protein kinase B (also known as Akt), and other pathways [6,26,27]. The dysregulated/sustained activation of receptor tyrosine kinase (RTK) signaling pathways is a frequent hallmark of glioblastoma. Hence, developing drugs that selectively target these pathways has been a commonly used strategy to control glioblastoma growth.

Studies have revealed that IL-8 (also known as CXCL8)/CXC chemokine receptor 1 and 2 (CXCR1/2, also known as IL8R1/2) signaling contributes to tumor cell proliferation, invasion, and neovascularization of glioblastoma [28]. However, little is known about transcription factors that regulate the expression of CXCR1/2. This study demonstrates for the first time a crucial role of DNA-PK/MYT1L interplay in transcriptional regulation of CXCR1, highlighting the regulatory role of the neuron-specific transcription factor MYT1L in a proliferative signaling loop involving CXC chemokine ligand 1 (CXCL1, also known as GROα) and IL-8/CXCR1 in glioblastoma that may represent a pharmacological target loop for therapeutic intervention for this disease.

## 2. Results

### 2.1. MYT1L Acts as an Oncogene in Glioblastoma Cells with Normal DNA-PK Activity

Our previous studies have shown that MYT1L is elevated in 46.9% of malignant glioma tissues (n = 32) and may contribute to the progression of glioblastoma through the interaction with the DNA-dependent protein kinase (DNA-PK) [29], highlighting a novel modulatory role of DNA-PK in MYT1L-dependent transcription. Here, we used a glioblastoma cell line (M059K) and a human embryonic kidney epithelial cell line (HEK293) (each having a normal DNA-PK function, see refs. [30,31]) as model systems to investigate the role of MYT1L in glioblastoma biology. The MTT assay showed that knockdown of MYT1L with a pool of three target-specific 19–25 nt siRNAs inhibited M059K cell proliferation (Figure 1A,B). MYT1L-siRNA also induced M059K cell apoptosis (Figure 1C) and a G2 cell cycle arrest (Figure 1D). To confirm these findings, we generated cell lines that stably express either GFP or MYT1L, including M059K-GFP, M059K-MYT1L, HEK293-GFP, and HEK293-MYT1L lines, and assessed their effect on cell growth. In both cell lines (M059K and HEK293), the enforced expression of MYT1L promoted cell proliferation (Figure 1E,F), accelerated S/G2 transition (Figure 1G,H), and attenuated apoptosis (Figure 1I,J); 13% apoptotic cells were also noted in the negative control siRNA group (Figure 1C).

To explore the mechanisms involved, we determined the activation of signaling pathways and the expression of key molecules associated with the regulation of proliferation, apoptosis, and cell cycle. Western blot analysis showed that in both cell lines, ectopic MYT1L caused an up-regulation of phosphorylated AKT1/2/3, AKT1, and cyclin D1 and a down-regulation of cyclin A2, CDK4, and caspase 3, whereas MYT1L had no effect on the expression of CDK6 (Figure 1K, Appendix A). Ectopic MYT1L down-regulated p21 expression in HEK293 cells and, likely, in M059K cells, although the band on the Western blot was too faint to be absolutely certain (Figure 1K, Appendix A). We also noted no changes in BCL2, CDK2, and ERK1/2 levels in M059K cells, while being significantly down-regulated HEK293 cells. Interestingly, certain differences were observed in the two cell lines in response to MYT1L expression. Ectopic MYT1L reduced the levels of phosphorylated ERK1/2, p27, and cyclin E1 in HEK293 cells but increased the levels in M059K cells. MYT1L induced BAX and cyclin D1 overexpression in HEK293 cells but suppressed their expression in M059K cells. Taken together, these results suggest that in glioblastoma cells with normal DNA-PK activity, MYT1L may serve as an oncogene that contributes to the progression of glioblastoma by activating ERK1/2 and AKT signaling pathways, inhibiting apoptosis and shortening S phase in the cell cycle.

### 2.2. MYT1L Inhibited Proliferation and Induced Apoptosis in Glioblastoma Cells with Loss of DNA-PK Function

Using the M059J cell line as a model system, we compared the role of MYT1L in DNA-PK-deficient glioblastoma cells and normal DNA-PK cells [30]. Surprisingly, the MTT assay showed that knockdown of MYT1L remarkably enhanced M059J cell proliferation (Figure 2A,B), attenuated apoptosis (Figure 2C), and accelerated S/G2 transition (Figure 2D). In contrast, the ectopic expression of MYT1L suppressed M059J cell proliferation (Figure 2E,F) and induced a G1 cell cycle arrest (Figure 2G), although it caused an inhibition in apoptosis (Figure 2H).

A Western blot analysis showed that ectopic MYT1L caused an increase in phosphorylated ERK1/2, phosphorylated AKT1/2/3, CDK2, cyclin D1, and cyclin E1. It also induced down-regulation of cyclin A2, CDK6, and p21 but had no effect on the expression of BAX, BCL2, and p27 (Figure 2I). Interestingly, the expression of caspase 3 and CDK4 was undetectable in either M059J-GFP or M059J-MYT1L cells (Figure 2I). Taken together, our findings suggest that MYT1L may function as a tumor suppressor that plays a pivotal role in the progression of glioblastoma cells with loss of DNA-PK function through the activation of ERK1/2 and AKT pathways and through the regulation of the expression of molecules associated with apoptosis and cell cycle regulation. This supports our previous report indicating a modulatory role of DNA-PK in MYT1L-regulated transcription.

### 2.3. DNA-PK and MYT1L Are Both Involved in CXCR1 Transcription

We hypothesized the existence of a proliferative signaling loop involving CXC chemokines (e.g., CXCL1 and CXCL8) and their receptors (e.g., CXCR1 and CXCR2) in glioblastoma growth. A study has indicated an important role of the IL-8-CXCR1/2 axes in glioblastoma cell proliferation, invasion, and neovascularization [28]. However, whether MYT1L is involved in the transcriptional regulation of CXCR1/2 remains unknown. Bioinformatics analysis predicted an MYT1L-binding sequence (5′-actcaagcgca-3′) within the CXCR1 promoter (Appendix A). The relationship between the MYT1L protein and CXCR1 mRNA levels was then examined in glioblastoma and neuroblastoma cell lines. Western blotting revealed that MYT1L was overexpressed in all three glioblastoma cell lines (A-172, M059J, and M059K) and in three of seven neuroblastoma cell lines that were examined (Figure 3A). Quantitative real-time RT-PCR (qRT-PCR) showed that the CXCR1 expression was up-regulated in A-172 and M059K glioblastoma cell lines (Figure 3B), which positively correlated with MYT1L protein levels. In neuroblastoma cell lines, CXCR1 was up-regulated in SH-SY5Y, SK-N-MC, and SK-N-SH lines (Figure 3B), whereas only SH-SY5Y and SK-N-MC lines displayed a positive correlation between the MYT1L protein and CXCR1 mRNA levels. These results indicate that CXCR1 was overexpressed in 66.7% (n = 6) of MYT1L-positive brain cancer cell lines examined, implicating a contributing role of MYT1L in CXCR1 transcription, despite the Cancer Genome Atlas (TCGA) data analysis that showed no correlation between the levels of MYT1L mRNA and CXCR1 mRNA in both glioblastoma (R^2^ = −0.016309) and neuroblastoma (R^2^ = −0.032546) (Appendix A). Next, we used the stable MYT1L-expressing HEK293, M059J, and M059K cell lines as a model system to further establish the relationship. The qRT-PCR analysis showed that ectopic MYT1L induced CXCR1 expression in cells with a normal DNA-PK activity (HEK293 and M059K) (Figure 1F and Figure 3C), whereas it suppressed CXCR1 expression in DNA-PK-deficient cells (M059J) (Figure 2F and Figure 3C). Western blot analysis indicated that CXCR1 was overexpressed in HEK293 and M059K cells in response to ectopic MYT1L (Figure 3D), which is consistent with its mRNA levels in these cell lines (Figure 2C). Although ectopic MYT1L had no effect on the level of CXCR1 protein in M059J cells (Figure 3D), it reduced the level of CXCR2 protein (Figure 3E). Interestingly, CXCR2 proteins were undetectable in HEK293 and M059K cells (Figure 3E).

To establish a mechanistic link between MYT1L and CXCR1, a 3.1 kb CXCR1 promoter/luciferase reporter construct harboring a candidate MYT1L consensus cis-acting element (−2114/−2103) was generated (Figure 3F, upper panel). We mutated this element to confirm the specific binding of MYT1L to this element in the CXCR1 promoter (Figure 3F, lower panel). The enforced MYT1L expression in the cells with normal DNA-PK significantly induced the luciferase activity driven by the wild-type CXCR1 promoter (wtCXCR1-Luc) (Figure 3G, top and middle panels). In contrast, when the ^−2111^CAA^−2109^ motif was mutated to ^−2111^TTT^−2109^ (mtCXCR1-Luc), the responsiveness of the MYT1L promoter was attenuated. The enforced MYT1L expression in the cells with loss of DNA-PK function reduced the activity of luciferase driven by the wild-type CXCR1 promoter (Figure 3G, bottom panel), whereas MYT1L responsiveness was attenuated in the mutant construct. Taken together, these results suggest that MYT1L is directly involved in the transcriptional control of CXCR1 expression, while DNA-PK appears to be essential for modulating the transcriptional activity of MYT1L.

### 2.4. IL-8- and GROα-CXCR1/ERK1/2 Proliferative Signaling in Glioblastoma Cells

Next, we looked at the contributing role of MYT1L to an IL-8- and GROα-CXCR1 signaling loop in glioblastoma cells. Global gene expression profiling of HEK293 cells that stably expressed either MYT1L or GFP identified two CXCR1 ligands, CXCL1 (GROα) and CXCL8 (IL-8), that were up-regulated in response to ectopic MYT1L expression (Appendix A). qRT-PCR showed that IL-8 was up-regulated in the HEK293 and M059J cell lines that ectopically expressed MYT1L (Figure 4A, left and right panels), whereas no significant alterations were found in the M059K cells (Figure 4A, middle panel, *p* = 0.0666). The qRT-PCR and immunofluorescence (IF) analyses indicated that the ectopic expression of MYT1L caused an overexpression of GROα in HEK293 cells at both mRNA and protein levels (Figure 4B,C, Appendix A), which supports our gene expression profiling data. Similar results were also found in M059J cells (Figure 4D,E, Appendix A). Interestingly, no change in the levels of GROα protein was detected in M059K cells (Figure 4G), although the enforced expression of MYT1L attenuated GROα mRNA levels (Figure 4F).

To see the significance of the IL-8- and GROα-CXCR1 signaling loop in glioblastoma, we determined the expression of GROα, IL-8, CXCR1/2, and the status of ERK1/2 and AKT pathways in 4 brain cancer cell lines. The qRT-PCR indicated that IL-8 was up-regulated in M059J, M059K, and SK-N-BE(2) cell lines, while it was down-regulated in the A-172 cell line (Figure 4H). Western blot analysis showed that GROα was overexpressed in M059J and M059K cell lines, whereas it was undetectable in SK-N-BE(2) and A-172 cell lines (Figure 4I). Although CXCR2 expression was elevated in all brain cancer cell lines examined, CXCR1 appeared to be more selectively overexpressed in M059J and M059K cell lines up-regulated in GROα and/or IL-8 (Figure 4I). A higher CXCR1 expression was also noted in the DNA-PK wild-type M059K line. Notably, the phosphorylated ERK1/2 and AKT1/2/3 were both up-regulated in all brain cancer cell lines measured (Figure 4I), implicating a contributing role of these two pathways in brain carcinogenesis.

To determine the intracellular pathway(s) triggered by GROα-CXCR1 and IL-8-CXCR1 binding, we knocked down CXCR1 in the M059K cells using 5 pooled target-specific siRNAs and looked at the effect on cell proliferation. Western blot analysis indicated that CXCR1 was profoundly knocked down by 80 nM CXCR1 siRNA, leading to a reduction in the phosphorylated ERK1/2 (Figure 4J) but leaving pAKR1/2/3 unaffected, consequently resulting in a suppression in cell proliferation (Figure 4K). Taken together, these results suggest that CXCR1 and its ligands GROα and IL-8 were aberrantly expressed in brain cancer cells and that CXCR1 mediated proliferative signaling that was triggered by its ligands via the ERK1/2 pathway.

## 3. Discussion

Cell surface receptors and their coupled intracellular signaling pathways have been intensively studied in many physiological and pathological processes over five decades. Due to the critical feature of these receptor-mediated signaling in promoting malignant proliferation, invasion, and metastasis, designing/seeking small molecules and monoclonal antibodies that target receptors and downstream pathways has become a pivotal strategy in anti-cancer drug discovery. Therefore, a better understanding of the key players in the signaling pathways may offer opportunities for therapeutic intervention.

This study reveals a positive feedback DNA-PK/MYT1L-CXCR1 proliferative signaling loop that contributes to the progression of glioblastoma via the ERK1/2 pathway and provides a novel insight into the role of DNA-PK in the MYT1L-mediated transactivation of CXCR1. MYT1L is predominantly expressed in human brains, in particular fetal brains [32]. Several key studies have demonstrated the pivotal role of MYT1L in the direct conversion of human non-neuronal cells to neurons [33,34,35]. However, the contributing role of MYT1L in carcinogenesis remains poorly understood. Our previous studies have uncovered that miR-141 was down-regulated in glioblastoma and was able to suppress cell proliferation by directly targeting MYT1L, implicating an oncogenic role of MYT1L in glioblastoma. In this study, we found that in normal DNA-PK glioblastoma M059K cell lines, knockdown of MYT1L attenuated cell proliferation and induced apoptosis and S-phase cell cycle arrest (Figure 1A–D). In contrast, the ectopic expression of MYT1L promoted cell proliferation, inhibited apoptosis, and shortened the S phase (Figure 1E,F,H,J). Similar phenotypic alterations were also noted in HEK293 cells in response to ectopic MYT1L (Figure 1E–G,I). These results indicated that in cooperation with DNA-PK, MYT1L may function as an oncogene in the progression of glioblastoma. However, in the loss-of-function DNA-PK M059J cell line, the knockdown of MYT1L promoted cell proliferation, attenuated apoptosis, and shortened the S phase (Figure 2A–D). Conversely, the enforced expression of MYT1L inhibited cell proliferation and induced a G1 cell cycle arrest (Figure 2E–G), suggesting a tumor suppressor role of MYT1L in DNA-PK-deficient glioblastoma cells, although it suppressed apoptosis (Figure 2H). Mechanically, the sustained activation of the AKT pathway in both HEK293 and M059K cells and the activation of the ERK1/2 pathway in M059K cells (Figure 1K) may contribute to cell proliferation induced by ectopic expression of MYT1L, which supports the key role of these two well-studied oncogenic signaling pathways in the development of human malignancies [36]. However, the proliferative inhibition mediated by ectopic expression of MYT1L in M059J cells may not be attributed to the activation of these two pathways (Figure 2I). A total of 13% apoptotic cells were also found in the negative control siRNA group (Figure 1C), which may reflect the cytotoxic effect of either transfection reagent (Lipofectamine 3000) or negative control siRNA.

The cell cycle is tightly regulated by cyclins, CDKs, and CDK inhibitors. Although cyclin E1 was down-regulated in HEK293, while it was up-regulated in M059K cells in response to MYT1L expression (Figure 1K), that may play a role in shortening the S phase (Figure 1G,H) because of the critical requirement of cyclin E in the S phase progression [37,38]. Furthermore, p21, a well-known CDK inhibitor, was down-regulated in HEK293 and M059K cells in response to MYT1L (Figure 1K), which may also contribute to shortening the S phase. Moreover, p21 has been shown to promote cell death in cancer cells via the activation of autophagy [39,40,41]; the down-regulated p21 may be a causal player in attenuating apoptosis in HEK293 cells that express MYT1L (Figure 1I,J). Interestingly, CDK6 was down-regulated in M059J cells in response to MYT1L (Figure 2I), which may play a role in a G1 cell cycle arrest (Figure 2G). Ectopic MYT1L induced down-regulation of caspase 3 (Figure 1K, Appendix A) in both HEK293 and M509K cells, which may play a contributing role in apoptotic inhibition (Figure 1I,J). However, this is not the case in M059J due to the absence of caspase 3 in both GFP- and MYT1L-M059J cells (Figure 2I).

As a transcription factor, the transcriptional targets of MYT1L, unfortunately, remain largely unknown. Here, we discovered that MYT1L could directly regulate the transcription of CXCR1 through a novel cis-acting element (−2114/−2103) at the CXCR1 promoter. We noted that the overexpression of MYT1L correlated with CXCR1 mRNA levels in 4 out of 6 brain cancer cell lines (Figure 3A,B) and the induction of CXCR1 in the normal DNA-PK HEK293 and M059K cells in response to the ectopically expressed MYT1L (Figure 3C,D), supporting the opinion that CXCR1 is a transcriptional target of MYT1L. Surprisingly, in M059J cells with loss of DNA-PK function (Appendix A), the enforced expression of MYT1L failed to induce CXCR1 transcription (Figure 3C), although the protein levels of CXCR1 were not consistent with its mRNA levels (Figure 3C,D). Taken together, our results demonstrate that MYT1L functionality depends on DNA-PK activity. Furthermore, the ectopically expressed MYT1L increased the luciferase activity of the wild-type CXCR1 promoter/luciferase reporter in HEK-293 and M059K cells with the normal expression of DNA-PK, which was abolished when the mutant construct was used (Figure 3G). However, the ectopic expression of MYT1L suppressed the luciferase activity in the DNA-PK-deficient M059J cells (Figure 3G). Moreover, the enforced expression of MYT1L resulted in the sustained activation of the AKT pathway in all three cell lines examined (Figure 1K and Figure 2I). The ERK1/2 pathway was activated in M059K and M059J cells, while it was inhibited in HEK293 cells in response to MYT1L (Figure 1K and Figure 2I). The activation of the ERK1/2 pathway was CXCR1-dependent, whereas the AKT pathway was not (Figure 4J). Based on these findings, we may propose that under non-stimulated conditions, MYT1L may act as a transcriptional repressor suppressing CXCR1 transcription; DNA-PK may function as a coactivator, and once it is phosphorylated by the CXCR1-dependent activation of the ERK1/2 kinase, it, in turn, phosphorylates MYT1L; the phosphorylated MYT1L recruits histone acetyltransferases (HATs) and transcription factors (TFs) to the CXCR1 promoter and promotes CXCR1 transcription (Figure 5). DNA-PK (also known as PRKDC and p350) is a nuclear serine/threonine protein kinase complex that is composed of a catalytic subunit of DNA-PKcs and a heterodimer of Ku proteins (Ku70/Ku80). Although one of the most well-defined functions of DNA-PK is to govern the repair of DNA double-strand breaks through both non-homologous end-joining (NHEJ) and homologous recombination (HR) [42], it was originally identified as a component of the Sp1 transcription complex in which it may modulate the transcriptional activity of the complex by phosphorylating Sp1 [43]. For the first time, our findings demonstrated that DNA-PK may functionally modulate the transcriptional activity of MYT1L by phosphorylation, eventually leading to the transcriptional activation of CXCR1, although this requires further validation using chromatin immunoprecipitation quantitative PCR (ChIP-qPCR) and the electrophoretic mobility shift assay (EMSA) when a ChIP-grade antibody to the phosphorylated MYT1L is commercially available. Our previous work demonstrated that DNA-PK may also function as a repressor of MYT1L, resulting in the transcriptional inhibition of tumor suppressor *p21*, a transcriptional target of MYT1L [29]. Several publications support the dual role of DNA-PK in the transcriptional control of gene expression. DNA-PK inhibited the transcription of the *xanthine oxidoreductase* gene via E-box/TATA-like elements [44]. Simultaneously, Ku proteins may also act as a transcriptional recycling coactivator of the androgen receptor [45].

Chemokines are a superfamily of small (about 8 to 14 kDa) cytokine-like proteins that selectively regulate the recruitment and trafficking of leukocytes to inflammatory sites through chemoattraction [46]. Members of this family have been divided into four subfamilies, CXC, CC, C, and CX3C, based on the arrangement of the first two of the four conserved cysteine residues in the amino terminus of the proteins, which can bind CXCR, CCR, XCR1, and CX3CR1, respectively. Chemokines play a critical role in inflammatory reactions. IL-8 is a well-characterized ELR+ CXC chemokine that attracts neutrophils through binding CXCR1 and CXCR2 receptors on the cell surface [47]. It is suspected that the signature of chemokines that persist at inflammatory sites may be important in the development of chronic diseases and can contribute to the development of malignancies [48]. There is sufficient evidence to show that chemokines also play a significant role in cancer, including melanoma [49], breast [50], colon [51], esophageal [52], prostate [53], and non-small cell lung [54] cancers, in addition to its role in the development of inflammatory responses. Here, we showed that IL-8 was significantly up-regulated in HEK293 and M059J cells, while it was only slightly increased in M059K cells in response to the ectopically expressed MYT1L (Figure 4A). Unlike IL-8, GROα more selectively binds to the CXCR2 receptor, while the affinity to CXCR1 is significantly lower [55]. Since the CXCR2 receptor was undetectable in HEK293 and M059K cells (Figure 3E), GROα in these cells could only function through CXCR1. The enforced expression of MYT1L caused an elevation of GROα in HEK293 and M059J cells (Figure 4B–E). Although GROα mRNA was down-regulated, its protein level was unaffected in M059K cells in response to the ectopically expressed MYT1L (Figure 4F,G). IL-8 and/or GROα bind to CXCR1, eventually promoting glioblastoma proliferation via the ERK1/2 pathway that can be blocked by CXCR1 siRNA (Figure 4J,K), suggesting that the activation of the proliferative ERK1/2 pathway in glioblastoma is, at least in part, CXCR1-dependent. Although this is the first report regarding the critical role of DNA-PK/MYT1L in the proliferative IL-8/GROα-CXCR1 signaling loop in glioblastoma, the significant contributing role of IL-8-CXCR1/2 axes in the proliferation and angiogenesis of glioblastoma cells has been demonstrated before [28]. Taken together, we propose a potential DNA-PK/MYT1L-CXCR1 signaling loop in the progression of glioblastoma (Figure 5).

We speculate that the DNA-PK/MYT1L-CXCR1-ERK1/2 might be up-regulated at WHO grade IV and/or recurrent glioblastomas due to chemo- or radio-resistance. DNA-PK is a key player in DNA damage response on double-strand breaks. Activation of DNA damage response facilitates glioma stem cells to develop radio-resistance [56]. Furthermore, DNA-PK is a master kinase of the proliferative/progenitor subtype of glioblastoma, guiding targeted cancer therapy [57]. Immunohistochemical staining has indicated the expression of IL-8 in 66.7–67.3% of WHO grade IV glioblastoma tissues, with the expression of its receptor CXCR1 found primarily in both tumor-associated vessels and grade IV glioblastomas [28], supporting the previous report showing a crucial role of IL-8/CXCR1/2 signaling in glioblastoma stem cells [58]. Moreover, the IL-8/CXCR1/STAT3 pathway is crucial for the maintenance of glioblastoma stem cells [59].

Glioblastoma and neuroblastoma are two common types of brain tumors that occur in different aged populations. Our previous studies indicated that MYT1L was overexpressed in glioblastoma cell lines and 46.9% of malignant glioma tissue samples (n = 32) [29], functioning as an oncogene in glioblastoma cells with normal DNA-PK activity. To see whether MYT1L is also up-regulated in neuroblastoma, we compared its expression in both glioblastoma and neuroblastoma cell lines. It was found that MYT1L was overexpressed in 3 out of 7 neuroblastoma cell lines examined (Figure 3A), suggesting that MYT1L up-regulation may be a common event in both glioblastoma and neuroblastoma. Although the SH-SY5Y cell line is a subclone of the SK-N-SH cell line, the global gene expression microarray showed a profound differential expression of genes in these two cell lines [60]. However, the mechanism involved is unclear. In the present study, we noted that MYT1L was up-regulated in the SH-SY5Y cell line at both mRNA and protein levels (Figure 3A,B), while in its parental line, SK-N-SH, only MYT1L mRNA was up-regulated. The MYT1L protein was undetectable, which may implicate the involvement of post-transcriptional regulation, such as miRNA(s), in MYT1L expression.

Some of the molecules analyzed in this manuscript can be molecular targets in cancer therapy. The IL-8-CXCR1/2 axis has been proposed as a potential therapeutic target for numerous cancers due to its contributing roles in proliferation, migration/invasion, angiogenesis, and tumor immunosuppression [61,62]. The tumor-produced IL-8 is a potent chemoattractant that recruits CXCR1/2-expressing human myeloid-derived suppressor cells to tumor foci, playing a crucial role in immune resistance [63]. IL-8 receptor CXCR1 may be a biomarker for cancer stem cells [64], including glioblastoma stem cells [58]. Targeting myeloid cell CXCR1/2 enhances antitumor immunity in pancreatic cancer [65]. Neutralizing IL-8 or inhibiting its receptor CXCR1/2 potentiates anti-PD-1-mediated antitumor immunotherapy for glioma [66]. These findings highlight IL-8 and CXCR1/2 as potential therapeutic targets for cancer. McClelland and colleagues have summarized numerous inhibitors developed to target IL-8 or its receptor CXCR1/2 for the treatment of chronic obstructive pulmonary disease (COPD), asthma, diabetes, pneumonia, and solid tumors, including prostate cancer [62]. Phase 1 and 2 clinical trials in prostate cancer with Navarixin, a small-molecule inhibitor of CXCR1/2, have been completed. Phase 1 and 2 clinical trials in prostate cancer with BMS-986253, a monoclonal antibody against IL-8, are still active. However, none of these inhibitors have yet been approved to treat cancer.

The ERK1/2 signaling pathway contributes to numerous biological and pathological processes. Excessive activation of the ERK1/2 pathway is a hallmark of human malignancies [67,68,69,70,71,72,73] due to its crucial role in maintaining sustained cellular proliferation, resistance to cell death, angiogenesis, invasion, and metastasis. Many FDA-approved drugs target upstream regulators of the ERK1/2 pathway, resulting in an indirect inhibition of ERK1/2 kinase [74]. ERK1/2 signaling may be reactivated upon the development of drug resistance. Liu and colleagues have described 10 small-molecule inhibitors of ERK1/2 which were developed recently and have undergone clinical trials in cancer patients [75]. Some of them have shown promising results in the treatment of cancers. Surprisingly, a large body of evidence also indicates a contributing role of ERK1/2 activation in cancer cell apoptosis primarily induced by chemical compounds, including glioblastoma, bladder, breast, colon, endometrial, head and neck, lung, renal cell, and testicular germ cell carcinomas [76,77,78,79,80,81,82,83,84,85], and mechanically, through activation of caspase 3 and induction of reactive oxygen species.

Both oncogene and non-oncogene addiction have been demonstrated as fantastic targets for cancer therapy due to their crucial role in supporting cancer progression [86]. Using a unique glioblastoma cell model system, M059J with deficient DNA-PK, while M059K with normal DNA-PK, we demonstrate in vitro that MYT1L-overexpressing cells show non-oncogene addiction to DNA-PK. Genomic instability due to defects in DNA repair genes is a key hallmark of cancer that drives the development of human malignancies [87]. However, these defects may also offer cancer cells opportunities to evolve a dependency on a non-oncogene addiction gene. For instance, BRCA2-deficient cancer cells are more dependent on PARP1 to repair DNA for survival, resulting in an increased sensitivity to PARP1 inhibitors [88], which kill the cancer cells through the genetic concept of synthetic lethality [89]. Targeting mediators of DNA repair has become a state-of-the-art strategy for cancer [90], leading to the development of inhibitors of key mediators of DNA repair, including DNA-PK. Interestingly, genetic and pharmacological targeting of a strong non-oncogene addiction to DNA-PKcs (DNA-dependent protein kinase catalytic subunit) leads to the accumulation of DNA double-strand breaks in ATM-defective cancer cells, which triggers cell apoptosis via the proapoptotic signaling pathway [91]. These results are supported by our findings showing the dependence of MYT1L-overexpressing glioblastoma cells on DNA-PK. VX-984 and M3814 are two selective DNA-PK inhibitors [92,93] that profoundly inhibit tumor growth in tumor xenograft models by enhancing radiotherapy. These two inhibitors are currently in ongoing clinical trials.

The DNA-PK/MYT1L-CXCR1-ERK1/2 proliferative signaling loop we proposed here was primarily based on data obtained from glioblastoma cell lines. It is interesting to look at the expression of DNA-PK, MYT1L, and CXCR1 and the levels of phosphorylated ERK1/2 in a large cohort of glioblastoma tissue samples and correlate the levels of these molecules to clinicopathological parameters of this disease. It is also interesting to look at the effect of MYT1L on tumor growth in an in vivo tumor xenograft animal model using GFP-/MYT1L-M059J and GFP-/MYT1L M059K cells, further confirming our in vitro findings.

## 4. Materials and Methods

### 4.1. Cell Culture

The schematic diagram of experimental procedures is shown in Appendix A. Human glioblastoma cell lines M059J, M059K, A172 and neuroblastoma cell lines SH-SY5Y and SK-N-BE(2), the embryonic kidney HEK293 cell line, and the foreskin fibroblast BJ-5ta cell line were purchased from ATCC (Manassas, VA, USA). Other neuroblastoma cell lines, IMR-5, IMR-32, SK-N-AS, SK-N-MC, and SK-N-SH, are gift lines kindly provided by Dr. Aru Narendran (Arnie Charbonneau Cancer Institute, University of Calgary, Calgary, AB, Canada). M059J (DNA-PK-deficient) and M059K cells (normal DNA-PK), originally isolated from a tumor specimen of a 33-year-old male patient with untreated malignant glioblastoma, were grown in a 1:1 mixture of Dulbecco’s Modified Eagle’s Medium (DMEM) and Ham’s F12 medium with 2.5 mM L-glutamine adjusted to contain 15 mM HEPES, 0.5 mM sodium pyruvate and 1.2 g/L sodium bicarbonate supplemented with 1% non-essential amino acids (NEAA), 10% fetal bovine serum (FBS) and 1% penicillin/streptomycin (P/S). A172 and BJ-5ta cells were grown in an ATCC-formulated DMEM supplemented with 10% FBS and 1% P/S. IMR-32, SK-N-MC, and SK-N-SH cells were grown in ATCC-formulated Eagle’s Minimum Essential Medium (EMEM) supplemented with 10% FBS and 1% P/S. SK-N-BE(2) and SH-SY5Y cells were grown in a 1:1 mixture of ATCC-formulated EMEM and F12 Medium supplemented with 10% FBS and 1% P/S. IMR-5 cells were grown in RPMI1640 medium supplemented with 2 mM L-Glutamine, 1% NEAA, 10% FBS, and 1% P/S. SK-N-AS cells were grown in DMEM medium supplemented with 1% NEAA, 10% FBS, and 1% PS. HEK293 cells were grown in DMEM/high glucose (Thermo Scientific, Waltham, MA, USA) supplemented with 10% FBS and 1% P/S. All cells were cultured at 37 °C in a humidified atmosphere of 5% CO_2_.

### 4.2. MTT Assay

M059J and M059K cells grown to 90% confluence were transiently transfected with either 40 nM MYT1L siRNA (Santa Cruz Biotechnology, Dallas, TX, USA) or 40 nM AllStars negative control siRNA (QIAGEN, Hilden, NRW, Germany) or 20 nM or 80 nM negative control-A (Santa Cruz Biotechnology) or CXCR1 siRNA (Santa Cruz Biotechnology, Dallas, TX, USA) using Lipofectamine 3000 (Invitrogen, Waltham, MA, USA) per the manufacturer’s instructions. Twenty-four hours after transfection, 3.0 × 10^3^ cells per well were plated in 96-well plates. The medium was changed on day 3 and every day after day 3. For HEK293, M059J, and M059K cell lines stably expressing either MYT1L or GFP (HEK293-GFP and HEK293-MYT1L, M059J-GFP and M059J-MYT1L, M059K-GFP and M059K-MYT1L), 3.0 × 10^3^ cells per well were also plated in 96-well plates. The 3-(4,5-Dimethylthiazol-2-yl)-2,5-diphenyl tetrazolium bromide (MTT) assays were performed using the Cell Proliferation Kit I (Roche Diagnostics GmbH, Mannheim, BW, Germany) according to the manufacturer’s instructions. The spectrophotometric absorbance of samples was measured at 595 nm using a microtiter plate reader (FLUOstar Omega, Ortenberg, HE, Germany).

### 4.3. Apoptosis and Cell Cycle Analyses

M059J and M059K cells grown to 90% confluence were transiently transfected with either 40 nM MYT1L siRNA (Santa Cruz Biotechnology) or 40 nM AllStars negative control siRNA (QIAGEN, Hilden, NRW, Germany) using Lipofectamine 3000 (Invitrogen, Waltham, MA, USA) per the manufacturer’s instructions. At 72 h after transfection, the cells were harvested for the apoptosis and cell cycle analyses, which were performed using a BD FACSCanto™ II Flow Cytometer (BD Biosciences, Franklin Lakes, NJ, USA) with a propidium iodide staining solution and a BD Pharmingen™ V-FITC Annexin Apoptosis Detection Kit II (BD Biosciences, Franklin Lakes, NJ, USA) per the manufacturer’s instructions. For the cell lines stably expressing MYT1L, the cell cycle and apoptosis analyses were performed at the University of Calgary using the Nuclear-ID^TM^ Red Cell Cycle Kit (GFP-certified, Enzo Life Sciences, Farmingdale, NY, USA) and Annexin V-Cy3 (BioVision, Milpitas, CA, USA), respectively, according to the manufacturers’ instructions. All the experiments were performed in duplicate (2 technical replicates) or triplicate (3 technical replicates).

### 4.4. Generation of Stable Cell Lines Expressing MYT1L

M059J, M059K, and HEK293 cells grown to 90% confluence were transfected with either pCMV6-AC-MYT1L or pCMV6-AC-GFP purchased from Origene using Lipofectamine 3000 (Invitrogen, Waltham, MA, USA) per the manufacturer’s instructions. At 24 h after transfection, the cells were treated with G418. After G418 selection, the positive cells were sorted twice at the University of Calgary.

### 4.5. Western Blot Analysis

The cells were rinsed twice with ice-cold PBS and scraped off the plate in radioimmunoprecipitation assay buffer (RIPA). The total protein lysate prepared from human adult brain normal tissues was purchased from BioChain (Newark, CA, USA) and served as a normal control for human cell lines. The whole cellular lysates (30–100 µg per sample) were electrophoresed via 10% SDS-PAGE and electrophoretically transferred to PVDF membranes (Amersham Hybond^TM^-P, GE Healthcare, Chicago, IL, USA) at 4 °C for 1.5 h. The blots were incubated for 1 h with 5% nonfat dry milk to block the nonspecific binding sites and subsequently incubated with polyclonal/monoclonal antibodies specific to MYT1L (Abnova, Taipei, China) or AKT1 (Abcam, Cambridge, GH, UK), CDK2 (Abcam, Cambridge, GH, UK), DNA-PKcs (Abcam, Cambridge, GH, UK), p21 (Abcam, Cambridge, GH, UK) or CDK4 (Cell Signaling Technology, Danvers, MA, USA), CDK6 (Cell Signaling Technology, Danvers, MA, USA), cyclin A2 (Cell Signaling Technology, Danvers, MA, USA), cyclin D1 (Cell Signaling Technology, Danvers, MA, USA), cyclin E1 (Cell Signaling Technology, Danvers, MA, USA), ERK1/2 (Cell Signaling Technology, Danvers, MA, USA), p27 (Cell Signaling Technology, Danvers, MA, USA), pERK1/2 (Cell Signaling Technology, Danvers, MA, USA) or BAX (Santa Cruz Biotechnology, Dallas, TX, USA), BCL2 (Santa Cruz Biotechnology, Dallas, TX, USA), GROα (Santa Cruz Biotechnology, Dallas, TX, USA), IL8RB (Santa Cruz Biotechnology, Dallas, TX, USA), pAKT1/2/3 (Santa Cruz Biotechnology, Dallas, TX, USA), or CXCR1 (LSBio, Lynnwood, WA, USA) at 4 °C overnight. Immunoreactivity was detected using a peroxidase-conjugated antibody and visualized using the ECL Plus Western Blotting Detection System (GE Healthcare, Chicago, IL, USA). The blots were stripped before reprobing with an antibody against actin (Abcam, Cambridge, GH, UK) or GAPDH (Santa Cruz Biotechnology, Dallas, TX, USA).

### 4.6. Whole-Genome Gene Expression Profiling

Gene expression profiling was carried out as described previously [94]. Briefly, total cellular RNA was isolated from HEK293 cells stably expressing MYT1L (HEK293-MYT1L) or GFP (HEK293-GFP) using an Illustra RNAspin mini kit (GE Healthcare Life Sciences, Marlborough, MA, USA) according to the manufacturer’s instructions. Quantification, purity, and integrity of the RNAs were determined using a NanoDrop 2000c spectrophotometer (Thermo Scientific, Walthan, MA, USA) and an Agilent 2100 bioanalyzer (Santa Clara). The RNA samples with RIN values of 7 or higher were used for further analysis. Library preparation, hybridization, detection, beadChip statistical analysis, and data processing were performed accordingly. Genes with logFC scores more than +0.65 or less than −0.65 and at least one expression signal ≥ 50 were selected.

### 4.7. Quantitative Real-Time RT-PCR (qRT-PCR)

Total RNA isolated from BJ-5ta, A-172, M059J, M059K, IMR-5, IMR-32, SH-SY5Y, SK-N-AS, SK-N-BE(2), SK-N-MC, SK-N-SH, HEK293-GFP, HEK293-MYT1L, M059J-GFP, M059J-MYT1L, M059K-GFP, and M059K-MYT1L cells was subjected to the qRT-PCR analysis using an iScript^TM^ Select cDNA Synthesis Kit (Bio-Rad, Hercules, CA, USA) and SsoFast^TM^ EvaGreen Supermix (Bio-Rad, Hercules, CA, USA) and primer sets specific to GROα or CXCR1 or IL-8 (Bio-Rad, Hercules, CA, USA) with a CFX96 Real-Time System (Bio-Rad, Hercules, CA, USA). The glyceraldehyde-3-phosphate dehydrogenase (GAPDH) was used as a loading control. All experiments for qRT-PCR were performed in triplicate, the data were analyzed using the comparative C_t_ method, and results were shown as fold induction of mRNA.

### 4.8. Immunofluorescence

Immunofluorescence staining was performed as described previously [95]. Briefly, M059J, M059K, and HEK293 cells expressing either MYT1L or GFP were cultured on glass coverslips for 24 h (60–75% confluency), fixed with 4% paraformaldehyde in PBS, and permeabilized with an ice-cold methanol. After a 1 h incubation in a blocking buffer, the cells were incubated in a 1:50 dilution of an anti-GROα monoclonal antibody in the blocking buffer at 4 °C overnight. After five washes in the blocking buffer, the cells were incubated with a 1:500 dilution of a 488-fluorescent conjugated secondary antibody for 2 h at room temperature and counterstained with Prolong gold mounting media (Molecular Probe, Eugene, OR, USA). Fluorescence was observed using an inverted ZEISS microscope on both the green (Alexa 488) and blue (DAPI) channels and analyzed in a blinded manner.

### 4.9. Bioinformatics Analysis

The MYT1L binding sites in the CXCR1 promoter were predicted using Zinc Finger Protein-DNA Scoring Form, a DNA binding site predictor for Cys_2_His_2_ zinc finger proteins (http://zf.princeton.edu/form.php, accessed on 14 March 2025). The Cancer Genome Atlas (TCGA) datasets were used to analyze a relationship between MYT1L expression and CXCR1 expression in glioblastoma and neuroblastoma.

### 4.10. Site-Directed Mutagenesis

The site-directed mutation of the predicted MYT1L-binding sites in the CXCR1 promoter was performed using a QuickChange II Site-Directed Mutagenesis Kit (Agilent Technologies, Santa Clara, CA, USA) according to the manufacturer’s instructions. The following primers were utilized to generate the site-directed mutants. For the predicted motif at −2114/−2103 (tctcaaggggg, primary), CXCR1_MT2_F1: 5′-CTT GGC CCT GGT CTT TCT TTT GGG GGT GTC TCA CAG GGG-3′, CXCR1_MT2_R1: 5′-CCC CTG TGA GAC ACC CCC AAA AGA AAG ACC AGG GCC AAG-3′. The mutant sequence was confirmed by the automated sequencing.

### 4.11. Transient Transfection and Luciferase Assay

A 3133 bp fragment of the human CXCR1 promoter was cloned to the pGL3-Basic vector (Promega, Madison, WI, USA) between Kpn I and Xho I to generate the pGL3-wtCXCR-luc reporter, which was performed by GenScript. HEK293, M059J, and M059K cells grown to 90% confluency in a 6-well plate were transiently cotransfected with 0.5 μg of either pGL3-Basic (an empty vector, Promega) or pGL3-wtCXCR1-luc or pGL3-mtCXCR1^−2114/−2103^-luc in combination with 1.0 μg of pCMV6-MYT1L (MYT1L-GFP, OriGene, Austin, TX, USA) and 5.0 ng of pRL-TK (Promega, Madison, WI, USA) using Lipofectamine 3000 (Invitrogen) according to the manufacturer’s instructions. Twenty-four hours after transfection, the cells were lysed and the relative luciferase activity was measured with the Dual-Luciferase Reporter Assay System (Promega, Madison, WI, USA) using a luminometer (FLUOstar Omega, Ortenberg, HE, Germany).

### 4.12. Statistical Analysis

The Student’s *t*-test was used to determine the statistical significance of differences in GROα expression, IL-8 expression, cell growth, apoptosis, cell cycle, and luciferase activity between groups. Luciferase activity was measured in duplicate, while other experiments were performed in triplicate. A value of *p* < 0.05 was considered statistically significant.

## 5. Conclusions

This work, for the first time, reveals that MYT1L functions as a transcription factor governing the expression of CXC chemokine receptor CXCR1. The CXCR1 signaling promotes proliferation, inhibits apoptosis, and shortens the S phase in DNA-PK^+^ glioblastoma cells via activation of the ERK1/2 pathway, which can be blocked by CXCR1 knockdown. The function of MYT1L is DNA-PK-dependent, highlighting a key role of DNA-PK modulating the transcriptional activity of MYT1L. Our findings have demonstrated a positive feedback DNA-PK/MYT1L-CXCR1-ERK1/2 proliferative signaling loop in glioblastoma cells and might have significant therapeutic implications. In the future, our studies can be extended by analyzing the effect of CXCR1 knockdown in M059J cells with impaired DNA-PK. This would allow us to understand the interlink between DNA-PK-MYT1L-CXCR1-pERK1/2.

## Figures and Tables

**Figure 1 ijms-26-04398-f001:**
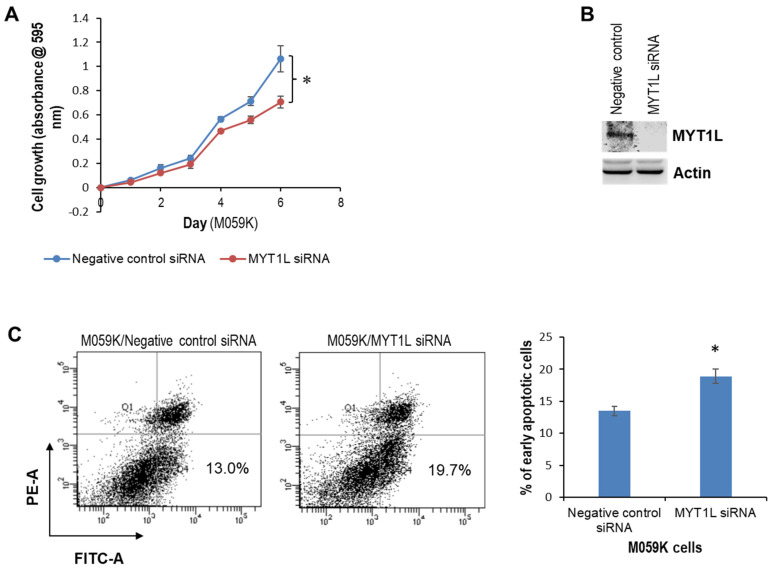
An oncogenic role of MYT1L in glioblastoma cells with normal DNA-PK activity. M059K cells grown to 90% confluency were transfected with either 40 nM MYT1L siRNA or 40 nM negative control siRNA. (**A**) At 24 h after transfection, cell growth was measured daily with the MTT assay kit in triplicate as described in “Methods”. (**B**) At 72 h after transfection, whole cellular lysates were prepared and subjected to Western blot analysis of MYT1L; actin served as a loading control. (**C**,**D**) At 72 h after transfection, cells were harvested for apoptosis (**C**) and cell cycle (**D**) analyses as described in “Methods”; all the analyses were performed in triplicate. (**E**) HEK293-MYT1L, HEK293-GFP, M059K-MYT1L, and M059K-GFP cells were replated in 96-well plates, and cell growth was measured daily with the MTT assay kit in triplicate as described in “Methods”. (**F**) Whole cellular lysates were prepared from HEK293-MYT1L, HEK293-GFP, M059K-MYT1L, and M059K-GFP cells and subjected to Western blot analysis of MYT1L; actin served as a loading control. (**G**–**J**) HEK293-MYT1L, HEK293-GFP, M059K-MYT1L, and M059K-GFP cells were harvested for cell cycle (**G**,**H**) and apoptosis (**I**,**J**) analyses as described in “Methods”; all the analyses were performed in triplicate. (**K**) Whole cellular lysates were prepared from HEK293-MYT1L, HEK293-GFP, M059K-MYT1L, and M059K-GFP cells and subjected to Western blot analysis of AKT1, BAX, BCL2, caspase 3, CDK2, CDK4, CDK6, cyclin A2, cyclin D1, cyclin E1, ERK1/2, p21, p27, pAKT1/2/3, and pERK1/2; actin served as a loading control. PE indicates fluorescent dye Propidium Iodide (PI) bound to DNA, FITC indicates FITC-Annexin V bound to cell membrane phospholipid phosphatidylserine. * indicates *p* < 0.05. Note: the whole cellular lysates used in “(**F**)” and “(**K**)” are exactly the same biological samples.

**Figure 2 ijms-26-04398-f002:**
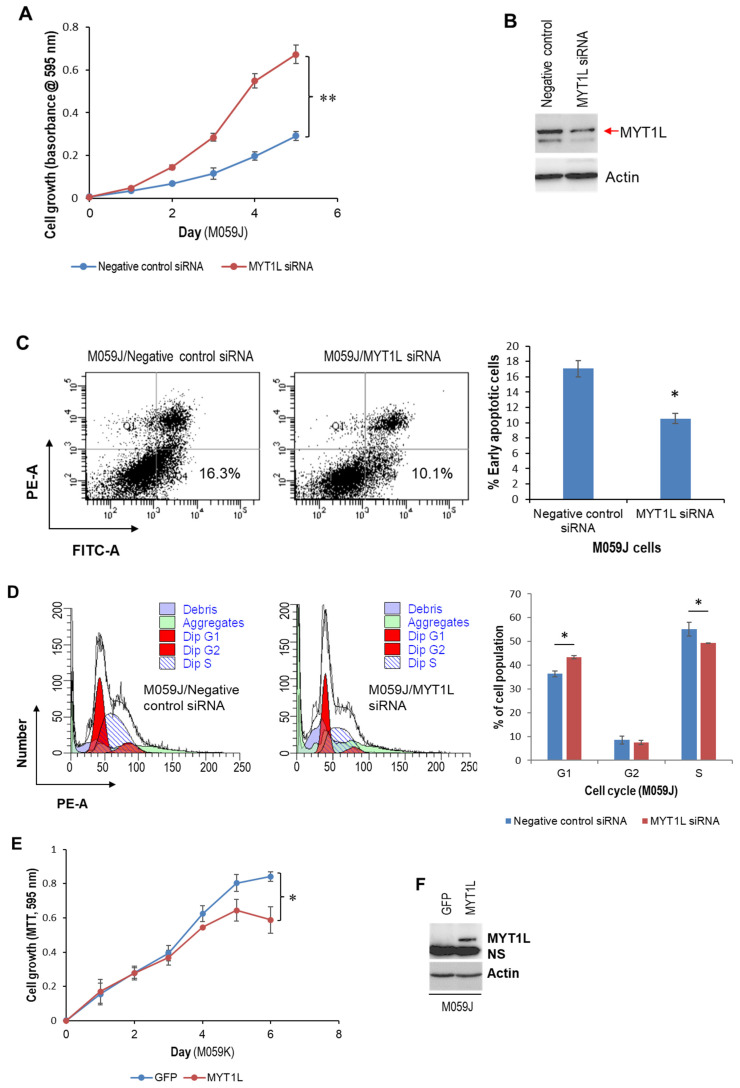
A tumor suppressive role of MYT1L in DNA-PK-deficient glioblastoma cells. M059J cells grown to 90% confluency were transfected with either 40 nM MYT1L siRNA or 40 nM negative control siRNA. (**A**) At 24 h after transfection. (**B**) Western blot analysis of MYT1L, 72 h after transfection; actin served as a loading control. (**C**,**D**) At 72 h after transfection, cells were harvested for apoptosis (**C**) and cell cycle (**D**) analyses as described in “Methods”; all the analyses were performed in triplicate. (**E**) M059J-MYT1L and M059J-GFP cells were replated in 96-well plates, and cell growth was measured daily with the MTT assay kit in triplicate as described in “Methods”. (**F**) Western blot analysis of MYT1L; actin served as a loading control. (**G**,**H**) M059J-MYT1L and M059J-GFP cells grown to subconfluency were harvested for cell cycle (**G**) and apoptosis (**H**) analyses. (**I**) Western blot analysis of AKT1, BAX, BCL2, caspase 3, CDK2, CDK4, CDK6, cyclin A2, cyclin D1, cyclin E1, ERK1/2, p21, p27, pAKT1/2/3, and pERK1/2; actin served as a loading control. PE indicates fluorescent dye Propidium Iodide (PI) bound to DNA, FITC indicates FITC-Annexin V bound to cell membrane phospholipid phosphatidylserine. * indicates *p* < 0.05; ** indicates *p* < 0.01. Note: the whole cellular lysates used in “(**F**)” and “(**I**)” are exactly the same biological samples.

**Figure 3 ijms-26-04398-f003:**
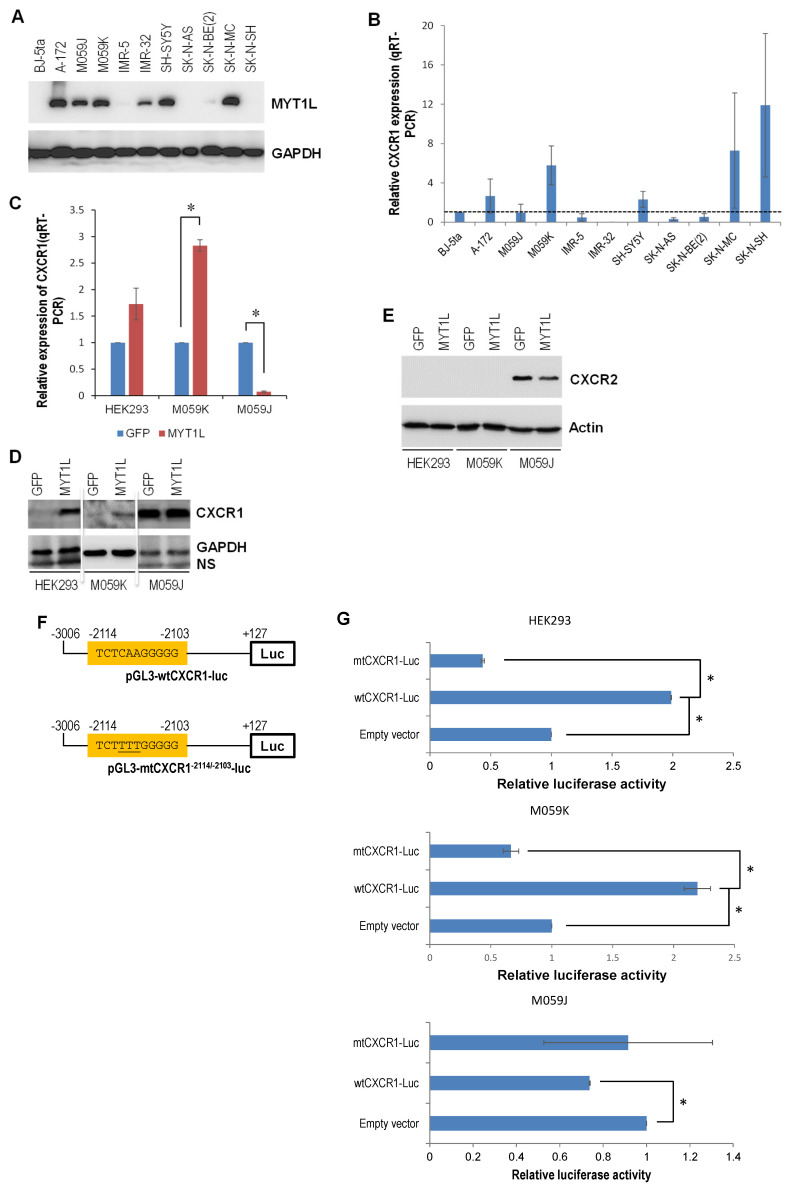
Transcriptional control of CXCR1 by MYT1L and DNA-PK. (**A**) Whole cellular lysates were prepared from BJ-5ta, A-172, M059J, M059K, IMR-5, IMR-32, SH-SY5Y, SK-N-AS, SK-N-BE(2), SK-N-MC, and SK-N-SH cells and subjected to Western blot analysis of MYT1L; GAPDH served as a loading control. (**B**) Total RNA was isolated from BJ-5ta, A-172, M059J, M059K, IMR-5, IMR-32, SH-SY5Y, SK-N-AS, SK-N-BE(2), SK-N-MC, and SK-N-SH cells and subjected to the qRT-PCR analysis of CXCR1 that was performed in triplicate. (**C**) Total RNA was isolated from HEK293-MYT1L, HEK293-GFP, M059J-MYT1L, M059J-GFP, M059K-MYT1L, and M059K-GFP cells and subjected to the qRT-PCR analysis of CXCR1 that was performed in triplicate. (**D**,**E**) Whole cellular lysates were prepared from HEK293-MYT1L, HEK293-GFP, M059J-MYT1L, M059J-GFP, M059K-MYT1L, and M059K-GFP cells and subjected to Western blot analysis of CXCR1 and CXCR2; actin and GAPDH served as a loading control. Note that the images of blots have been cropped and aligned in the same order as panels “(**C**)” and “(**E**)”. (**F**) A diagram of wild-type and mutant CXCR1 promoter/reporter constructs. (**G**) HEK293, M069J, and M059K cells were cotransfected with either an empty vector or pGL3-wtCXCR1-luc or pGL3-mtCXCR1^−2114/−2103^-luc reporter plasmid in combination with pCMV6-MYT1L and pRL-TK; at 24 h after transfection, the luciferase activity was measured in duplicate as described in “Methods”. * indicates *p* < 0.05.

**Figure 4 ijms-26-04398-f004:**
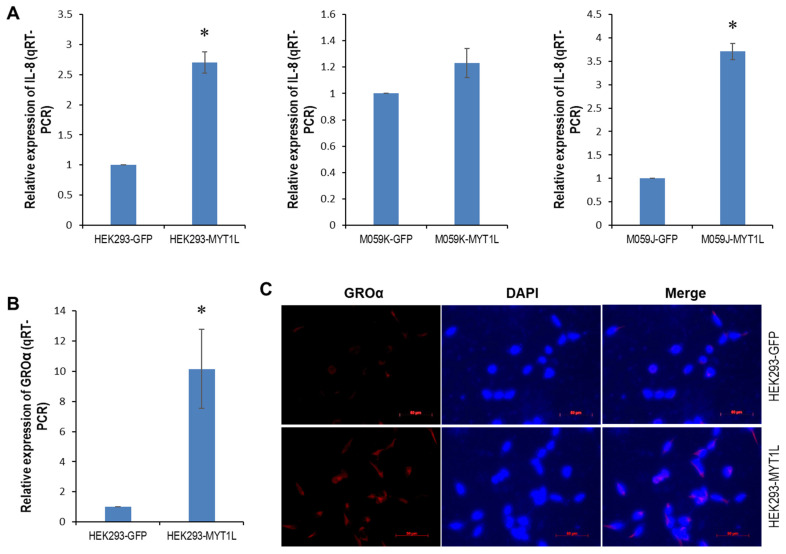
GROα- and IL-8-CXCR1/ERK1/2 signaling in glioblastoma. (**A**) Total RNA was isolated from HEK293-GFP (Passage# 3), HEK293-MYT1L (Passage# 5), M059K-GFP (Passage# 4), M059K-MYT1L (Passage# 6), M059J-GFP (Passage# 5), and M059J-MYT1L (Passage# 5) cells and subjected to the qRT-PCR analysis of IL-8 that was performed in triplicate. (**B**,**D**,**F**) Total RNA was isolated from HEK293-GFP, HEK293-MYT1L, M059K-GFP, M059K-MYT1L, M059J-GFP, and M059J-MYT1L cells and subjected to the qRT-PCR analysis of GROα that was performed in triplicate. (**C**,**E**,**G**) HEK293-GFP, HEK293-MYT1L, M059K-GFP, M059K-MYT1L, M059J-GFP, and M059J-MYT1L cells grown on a glass cover slide were subjected to immunofluorescence staining for GROα as described in “Methods”. (**H**) Total RNA was isolated from the brain normal tissue, M059J, M059K, SK-N-BE(2), and A-172 cells and subjected to the qRT-PCR analysis of IL-8 that was performed in triplicate. (**I**) Whole cellular lysates were prepared from the brain normal tissue, M059J, M059K, and A-172 cells and subjected to Western blot analysis of GROα, CXCR1, CXCR2, pERK1/2, ERK1/2, pAKT1/2/3, and AKT1; GAPDH served as a loading control. (**J**) M059K cells were transfected with 20 nM or 80 nM negative control-A or CXCR1 siRNA; at 48 h after transfection, whole cellular lysates were prepared and subjected to Western blot analysis of CXCR1, pERK1/2, ERK1/2, pAKT1/2/3, and AKT1; GAPDH served as a loading control. (**K**) M059K cells were transfected with either 80 nM negative control-A or 80 nM CXCR1 siRNA; at 24 h after transfection, cell growth was measured with the MTT assay kit in triplicate as described in “Methods”. * indicates *p* < 0.05.

**Figure 5 ijms-26-04398-f005:**
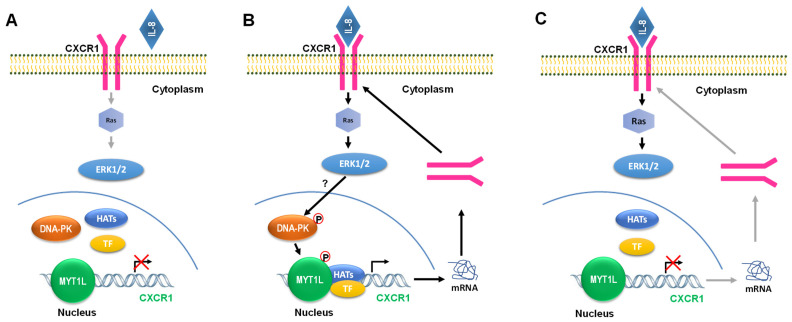
DNAPK/MYT1L-CXCR1 signaling loop in glioblastoma. (**A**,**B**) In glioblastoma cells with normal DNAPK activity, under non-stimulated conditions, the ERK1/2 pathway is inactivated, tumor suppressor MYT1L binds to CXCR1 promoter and acts as a repressor that blocks CXCR1 transcription (**A**); once ERK1/2 pathway activated by IL-8/GROα binding to receptor CXCR1, the activated ERK1/2 kinase may phosphorylate coactivator DNAPK that, in turn, phosphorylates MYT1L, the phosphorylated MYT1L recruits histone acetyltransferases (HATs) and transcription factors (TFs) to CXCR1 promoter, eventually resulting in CXCR1 transcription (**B**). (**C**) In glioblastoma cells with loss-of-function DNAPK, transcriptional repressor MYT1L can not be activated by DNAPK-dependent phosphorylation; as a result, the transcription of CXCR1 is reduced.

## Data Availability

The data generated in this study are publicly available at NIH National Library of Medicine, Bioproject ID: PRJNA846274.

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
