# Peer review of "A Positive Feedback DNA-PK/MYT1L-CXCR1-ERK1/2 Proliferative Signaling Loop in Glioblastoma"

_ijms, 2025, doi:10.3390/ijms26094398_

Round 1
Reviewer 1 Report
Comments and Suggestions for Authors
The manuscript by Wang, B. et.al., shows the apparent participation of the transcriptional proliferation loop (sic) MYTIL acting as a potential oncogene participating in some malignant features (P3,p1,L96-104 and P21,p3,L527-537) of Glioblastoma cells.
The experiments described in the text were adequately made and the findings would be interesting for future research of therapeutic attempts in the design of molecular genes against GBM cells; however, some points should be addressed.
- The text is very long to describe the findings of this transcription factor, it could be shortened to clarify relevant features.
- Figures are abundant and many are of poor quality (e.g. Figure 2I, 4I, 3,etc). The text suggests the description of Fig.IK as an explanation, it shows only a line of a blot analyses. Most figures could be eliminated and briefly described in the text.
- I found only Fig. 5 useful according to the text.
- In my opinion the potential use of this study was the features found of this transcriptional factor in a malignant brain tumor which could provide ideas for therapeutic investigations to be made against this very somber malignant brain tumor (P1,p2,L37-42). So, the text should feature the molecular relevance of MYT1L in GBM cells.
- The text on P6,L129-130 should be clarified.
- Transfection experiments could be described in general, rather than detailed context (P9,L162-179).
Author Response
Reviewer #1 comments
The manuscript by Wang, B. et.al., shows the apparent participation of the transcriptional proliferation loop (sic) MYTIL acting as a potential oncogene participating in some malignant features (P3,p1,L96-104 and P21,p3,L527-537) of Glioblastoma cells.
The experiments described in the text were adequately made and the findings would be interesting for future research of therapeutic attempts in the design of molecular genes against GBM cells; however, some points should be addressed.
- The text is very long to describe the findings of this transcription factor, it could be shortened to clarify relevant features.
Answer: We agree with your opinion, and have reduced the text a bit. But considering the readers may not be all experts in the field, we believe we need to be fairly comprehensive.
- Figures are abundant and many are of poor quality (e.g. Figure 2I, 4I, 3,etc). The text suggests the description of Fig.IK as an explanation, it shows only a line of a blot analyses. Most figures could be eliminated and briefly described in the text.
Answer: We partially agree. Some Western blotting images are not of perfect quality due to the poor quality of the antibodies, but good enough for publication. Eliminating some figures from the paper would likely make the paper look tidier, but it may prevent complete appreciation of the results and conclusions.
- I found only Fig. 5 useful according to the text.
Answer: Correct, but this figure would not be possible without showing all data sets, and thus all figures.
- In my opinion the potential use of this study was the features found of this transcriptional factor in a malignant brain tumor which could provide ideas for therapeutic investigations to be made against this very somber malignant brain tumor (P1,p2, L37-42). So, the text should feature the molecular relevance of MYT1L in GBM cells.
Answer: The molecular relevance of MYT1L in GBM cells has been discussed in details in our previous studies (see reference below).
Reference: Wang B, Li D, Yao Y, Heyns M, Kovalchuk A, Ilnytskyy Y, et al. The crucial role of DNA-dependent protein kinase and myelin transcription factor 1-like protein in the miR-141 tumor suppressor network. Cell Cycle 2019;18:2876-92.
- The text on P6,L129-130 should be clarified.
Answer: Sorry for the confusion. That has been clarified in the new version of the paper.
- Transfection experiments could be described in general, rather than detailed context (P9,L162-179).
Answer: We reduced the text of the figure, but since the figure has multiple components, we had to explain each panel.
Reviewer 2 Report
Comments and Suggestions for Authors
The authors of, "A Positive Feedback DNA-PK/MYT1L-CXCR1-ERK1/2 Proliferative Signaling Loop in Glioblastoma" have performed in vitro experiments to help define the role of the transcription factor MYT1L in regulation of CXCR1 in glioblastoma, with an approach of potentially identifying new therapeutic targets for glioblastoma. In addition to in vitro experiments performed by the authors, the TCGA database was also interrogated for glioblastoma and neuroblastoma patient data to bolster the significance of the findings. The in vitro experiments were performed using standard methods which were adequately described. The data are clearly presented. The summative figure 5 was very helpful for readers to appreciate the overall findings.
What is lacking in the manuscript is the rationale for comparing glioblastoma and neuroblastoma, beyond the simple idea that both originate in the brain. This rationale should be expanded and added to the manuscript. Another minor point is that the SH-SY5Y cell line is a subclone of the SK-N-SH cell line and therefore the authors may wish to discuss their results with those 2 cell lines in that context.
The authors should also revisit the introduction and discussion with respect to the references. Although valid and accurately cited, many of the references that the authors have cited are outdated. For example, the citation describing the incidence and prognosis of GBM patients is from 2011. Throughout the manuscript, the authors use the term 'recent' or 'recently' (line 78, from 2018; line 193, from 2014; line 321, from 2011) for references that although they are solid citations, they are not recent. These issues, although minor, detract from the data and conclusions of the manuscript.
Author Response
Reviewer #2 comments
The authors of, "A Positive Feedback DNA-PK/MYT1L-CXCR1-ERK1/2 Proliferative Signaling Loop in Glioblastoma" have performed in vitro experiments to help define the role of the transcription factor MYT1L in regulation of CXCR1 in glioblastoma, with an approach of potentially identifying new therapeutic targets for glioblastoma. In addition to in vitro experiments performed by the authors, the TCGA database was also interrogated for glioblastoma and neuroblastoma patient data to bolster the significance of the findings. The in vitro experiments were performed using standard methods which were adequately described. The data are clearly presented. The summative figure 5 was very helpful for readers to appreciate the overall findings.
What is lacking in the manuscript is the rationale for comparing glioblastoma and neuroblastoma, beyond the simple idea that both originate in the brain. This rationale should be expanded and added to the manuscript. Another minor point is that the SH-SY5Y cell line is a subclone of the SK-N-SH cell line and therefore the authors may wish to discuss their results with those 2 cell lines in that context.
Answer: We agree. Glioblastoma and neuroblastoma are two common types of brain tumor, which occur in different aged populations. Our previous studies indicated that MYT1L was overexpressed in glioblastoma cell lines and 46.9% malignant glioma tissue samples (n=32) (see reference below, Wang B et al, 2019), functioning as an oncogene in glioblastoma cells with normal DNA-PK activity. To see whether MYT1L is also upregulated in neuroblastoma, we compared its expression in both glioblastoma and neuroblastoma cell lines. It was found that MYT1L was overexpressed in 3 out of 7 neuroblastoma cell lines examined (Figure 3A), suggesting that MYT1L upregulation may be a common event in both glioblastoma and neuroblastoma. Although SH-SY5Y cell line is a subclone of the SK-N-SH cell line, global gene expression microarray showed a profound differential expression of genes in these two cell lines (see reference below, Looyenga BD et al, 2013). However, the mechanism involved is unclear. In the present study, we noted that MYT1L was upregulated in SH-SY5Y cell line at both mRNA and protein levels (Figure 3A and 3B), while in its parental line SK-N-SH, only MYT1L mRNA was upregulated, MYT1L protein was undetectable, that may implicate the involvement of post-transcriptional regulation, such as miRNA(s), in MYT1L expression. This information has been added in the “Discussion” of the revised paper.
Wang B, Li D, Yao Y, et al. The crucial role of DNA-dependent protein kinase and myelin transcription factor 1-like protein in the miR-141 tumor suppressor network. Cell Cycle 2019; 18(21):2876-2892.
Looyenga BD, Resau J, MacKeigan JP. Cytokine receptor-like factor 1 (CRLF1) protects against 6-hydroxydopamine toxicity independent of the gp130/JAK signaling pathway. PLoS ONE 2013; 8(6):e66548.
The authors should also revisit the introduction and discussion with respect to the references. Although valid and accurately cited, many of the references that the authors have cited are outdated. For example, the citation describing the incidence and prognosis of GBM patients is from 2011. Throughout the manuscript, the authors use the term 'recent' or 'recently' (line 78, from 2018; line 193, from 2014; line 321, from 2011) for references that although they are solid citations, they are not recent. These issues, although minor, detract from the data and conclusions of the manuscript.
Answer: We agree. The outdated citation (2011) describing the incidence and prognosis of GBM patients has been updated (2025) in the revised version. The term 'recent' or 'recently' (line 78, from 2018; line 193, from 2014; line 321, from 2011) has been deleted in the revised version.
Reviewer 3 Report
Comments and Suggestions for Authors
Dear Authors,
Your manuscript on the positive feedback loop of DNA-PK/MYT1L-CXCR1-ERK1/2 in glioblastoma is an interesting research approach. It is very comprehensive and shows an interesting new perspective. Nevertheless, I do have some points that may be worth considering:
The cell lines you use are not very suitable for glioblastoma research. As mentioned in the manuscript, the cell lines M059J and M059K are from the same origin, a 33-year-old patient. This raises some doubts about the correct diagnosis of glioblastoma, as glioblastoma are very rare in this age group. Therefore, it is absolutely necessary to prove the identity of theses cell lines. In addition, the reason for the difference of K and J should be disclosed.
The data you show of the MTT assay (Figure 1 A, E; Figure 2 A, E) has a strange bent after day 3. I speculate this is due to medium change that should be done after 3 days, but it is not described in the material and methods section. Also, the described procedures for other cell culture methods are probably incomplete, as the transfection is done at 90% confluency there is no more space for the cells to grow afterwards. If this is the case, the measured apoptosis or cell cycle changes may be attributed to overgrowth of the wells as well.
You analyse the proposed feedback loop from various sides which gives us a comprehensive view on these mechanisms, but I am missing a more robust validation of your hypothesis. In the discussion you mention that there are several inhibitors available for the different targets you show. I would recommend to use those in your experiments as well, to validate your findings with an independent method.
Minor remarks:
Line 63-65 You mention the key role of PDGFRB and a mouse model, but don’t describe why / how this key role is shown in this mouse model.
Line 97 The reference to Figure 1 B does not fit in here, as this is no MTT assay data, but the validation of the knock-down.
Line 117 you first mention that the measurements are done in triplicate. There are no further details in the Material and Methods section, so I recommend to specify this in more depth. Are these biological replicates and if so, how many technical replicates are done on one plate?
Line 439-445 is purely speculative and in respect to the tumor, totally wrong. By definition of WHO 2021 there are no glioblastoma with IDH mutation. The used reference is outdated by two new editions of brain tumor classification by the WHO (2016 and 2021).
Line 447, “advanced” glioblastomas are not defined by the WHO. You probably mean recurrent tumors after therapy.
Best regards
Author Response
Your manuscript on the positive feedback loop of DNA-PK/MYT1L-CXCR1-ERK1/2 in glioblastoma is an interesting research approach. It is very comprehensive and shows an interesting new perspective. Nevertheless, I do have some points that may be worth considering:
The cell lines you use are not very suitable for glioblastoma research. As mentioned in the manuscript, the cell lines M059J and M059K are from the same origin, a 33-year-old patient. This raises some doubts about the correct diagnosis of glioblastoma, as glioblastoma are very rare in this age group. Therefore, it is absolutely necessary to prove the identity of theses cell lines. In addition, the reason for the difference of K and J should be disclosed.
Answer: We agree. However, since these are commercially available cell lines, we think that proving the identity of these cell lines should be the seller’s responsibility, not researcher’s. These cell lines were purchased from ATCC, and the information on the short tandem repeat profiling for these cell lines was not available.
The data you show of the MTT assay (Figure 1 A, E; Figure 2 A, E) has a strange bent after day 3. I speculate this is due to medium change that should be done after 3 days, but it is not described in the material and methods section. Also, the described procedures for other cell culture methods are probably incomplete, as the transfection is done at 90% confluency there is no more space for the cells to grow afterwards. If this is the case, the measured apoptosis or cell cycle changes may be attributed to overgrowth of the wells as well.
Answer: Sorry for the confusion. Medium was changed on day 3, and every day after day 3, this information has been added in “Materials and methods” (see revised version). The transfection was done in this study using Lipofectamine 3000 (Invitrogen). Although Invitrogen announces that this transfection reagent “Improved cell viability – gentle on your cells, with low toxicity”, in our case, M509J and M059K cells are bit sensitive to Lipofectamine 3000. In transient transfection experiments, negative control siRNA could cause a 15-17% of apoptosis in these two cell lines (Figure 1C and 2C). 90% confluency won’t result in an overgrowth of these cells.
You analyse the proposed feedback loop from various sides which gives us a comprehensive view on these mechanisms, but I am missing a more robust validation of your hypothesis. In the discussion you mention that there are several inhibitors available for the different targets you show. I would recommend to use those in your experiments as well, to validate your findings with an independent method.
Answer: That is a great idea. We will test this in the future studies.
Minor remarks:
Line 63-65 You mention the key role of PDGFRB and a mouse model, but don’t describe why / how this key role is shown in this mouse model.
Answer: We agree. In this mouse model, the authors found that the retroviral vector expressing PDGFB drives primarily the development of glioblastoma which consistently expresses nestin protein, a well-known biomarker for neuroglial progenitor cells, indicating that activation of PDGFB signaling may be an initial or early event in neuro-oncogenesis. This information has been added in the revised version of the paper.
Line 97 The reference to Figure 1 B does not fit in here, as this is no MTT assay data, but the validation of the knock-down.
Answer: We respectfully disagree. Because the Figure 1A showed the inhibition of cell proliferation induced by MYT1L-knockdown, the validation of MYT1L-knockdown is essential reference here, otherwise it is hard to believe that the proliferative inhibition is caused by MYT1L-knockdown.
Line 117 you first mention that the measurements are done in triplicate. There are no further details in the Material and Methods section, so I recommend to specify this in more depth. Are these biological replicates and if so, how many technical replicates are done on one plate?
Answer: Sorry about the confusion. Three technical replicates were done on one plate. The related information has been added in revised “Materials and methods”.
Line 439-445 is purely speculative and in respect to the tumor, totally wrong. By definition of WHO 2021 there are no glioblastoma with IDH mutation. The used reference is outdated by two new editions of brain tumor classification by the WHO (2016 and 2021).
Answer: We respectfully disagree. The updated WHO classification 2016 classifies two major types of glioblastoma based on IDH mutations (see reference below, Le Rhun E et al, 2019). Although majority of glioblastomas (90%) are IDH-wildtype, small portion of glioblastomas are IDH-mutant, which is somehow supported by recent reports. To characterize the diverse landscape of glioma-associated myeloid cells in glioma, the authors purified CD45+CD3- immune cells with cell sorting from 21 resected IDH-wildtype glioblastoma,…and 2 IDH-mutant grade 4 astrocytomas (see reference below, Jackson C et al, 2025).
However, we deleted lines 439-445 to prevent confusion.
Le Rhun E, Preusser M, Roth P, et al. Molecular targeted therapy of glioblastoma. Cancer Treat Rev 2019; 80:101896.
Jackson C, Cherry C, Bom S, et al. Distinct myeloid-derived suppressor cell populations in human glioblastoma. Science 2025; 387(6731):eabm5214.
Line 447, “advanced” glioblastomas are not defined by the WHO. You probably mean recurrent tumors after therapy.
Answer: Sorry for the confusion. The “advanced” here means “recurrent”. That has been changed in the revised version.
Round 2
Reviewer 1 Report
Comments and Suggestions for Authors
No additional comments.